# Implementing quantum dimensionality reduction for non-Markovian stochastic simulation

Kang-Da Wu [1,2], Chengran Yang [3] ✉, Ren-Dong He[1,2], Mile Gu [3,4,5] ✉, Guo-Yong Xiang [1,2,6] ✉, Chuan-Feng Li [1,2,6], Guang-Can Guo[1,2,6] & Thomas J. Elliott [7,8,9] ✉

Complex systems are embedded in our everyday experience. Stochastic modelling enables us to understand and predict the behaviour of such systems, cementing its utility across the quantitative sciences. Accurate models of highly non-Markovian processes – where the future behaviour depends on events that happened far in the past – must track copious amounts of information about past observations, requiring high-dimensional memories. Quantum technologies can ameliorate this cost, allowing models of the same processes with lower memory dimension than corresponding classical models. Here we implement such memory-efficient quantum models for a family of non-Markovian processes using a photonic setup. We show that with a single qubit of memory our implemented quantum models can attain higher precision than possible with any classical model of the same memory dimension. This heralds a key step towards applying quantum technologies in complex systems modelling.

From chemical reactions to financial markets, and meteorological systems to galaxy formation, we are surrounded by complex processes at all scales. Faced with such rich complexity, we often turn to stochastic modelling to predict the future behaviour of these processes. Often, these future behaviours—and thus our predictions—are based not only on what we can observe about the current state of the process but also its past: they are *non-Markovian*.

To simulate such processes, our models must have a memory to store information about the past. Storing all past observations comes with a prohibitively large memory cost, forcing a more parsimonious approach to be adopted whereby we seek to distil the useful information from the past observations and store only this. Yet, when

processes are highly non-Markovian, we must typically retain information about observations far into the past, which still bears high memory costs. In practice, this leads to a bottleneck, where we trade-off reductions in the amount of past information stored against a loss in predictive accuracy.

Quantum technologies can offer a significant advantage in this endeavour, even when modelling processes with purely classical dynamics. They capitalise on the potential to encode past information into non-orthogonal quantum states to push memory costs below classical limits[1,2]. This advantage can be particularly pronounced for highly non-Markovian processes where the separation between quantum and classical memory costs can grow without bound[3–5].

[1]CAS Key Laboratory of Quantum Information, University of Science and Technology of China, Hefei 230026, People's Republic of China. [2]CAS Center For Excellence in Quantum Information and Quantum Physics, University of Science and Technology of China, Hefei 230026, People's Republic of China. [3]Centre for Quantum Technologies, National University of Singapore, 3 Science Drive 2, Singapore 117543, Singapore. [4]Nanyang Quantum Hub, School of Physical and Mathematical Sciences, Nanyang Technological University, Singapore 637371, Singapore. [5]MajuLab, CNRS-UNS-NUS-NTU International Joint Research Unit, UMI 3654, Singapore 117543, Singapore. [6]Hefei National Laboratory, University of Science and Technology of China, Hefei 230088, People's Republic of China. [7]Department of Physics & Astronomy, University of Manchester, Manchester M13 9PL, UK. [8]Department of Mathematics, University of Manchester, Manchester M13 9PL, UK. [9]Department of Mathematics, Imperial College London, London SW7 2AZ, UK. ✉e-mail: yangchengran92@gmail.com; mgu@quantumcomplexity.org; gyxiang@ustc.edu.cn; physics@tjelliott.net

Here, we experimentally realise quantum models for a family of non-Markovian stochastic processes within a photonic system. This family of processes has a tunable parameter that controls their effective memory length, and the memory dimension of the minimal classical model grows with the value of this parameter. Our quantum models can simulate any process within the family with only a single qubit of memory. Moreover, we show that even with the experimental noise in our implementation, our models are more accurate than any distorted classical compression to a single bit of memory. This is a significant advance over previous demonstrations of dimension reduction in quantum models, which were limited to models of Markovian processes[6], and so did not require the preservation of information in memory across multiple timesteps. Altogether, our work presents a key step towards demonstrating the scalability and robustness of such quantum memory advantages.

## Results

### Framework and theory

Stochastic processes consist of a series of (possibly correlated) random events occurring in sequence. Here, we consider discrete-time stochastic processes[7], such that events occur at regular time steps. The sequence of events can be partitioned into a past $\overleftarrow{x}$ detailing events that have already happened and a future $\overrightarrow{x}$ containing those yet to occur. Stochastic modelling then consists of sequentially drawing samples of future events from the process given the observed past.

This requires a model that can sample from the conditional form of the process distribution using a memory that stores relevant information from past observations. An (impractical) brute force approach would require the model to store the full sequence of past observations. A more effective model consists of an encoding function that maps from the set of pasts to a set of memory states $\{s_j\}$, and an evolution procedure that produces the next output (drawn according to the conditional distribution) and updates the memory state accordingly[8]. A more technical exposition is provided in the Methods.

A natural way to quantify the memory cost is in terms of the requisite size (i.e., dimension):

*Definition:* (Memory Cost) The memory cost $D$ of a model is given by the logarithm of the memory dimension, i.e., $D := \log_2 \dim(\{s_j\})$.

The number of (qu)bits required by the model's memory system corresponds to the ceiling of this quantity. For classical models, where the memory states must all be orthogonal, the memory cost is simply given by the (logarithm of the) number of memory states, i.e., $D = \log_2 |\{s_j\}|$. Moreover, when statistically exact sampling of the future is required, a systematic prescription for encoding the memory states with provably minimal classical memory cost is known—two pasts $\overleftarrow{x}$ and $\overleftarrow{x}'$ are mapped to the same memory state if and only if they give rise to the same conditional future statistics. These memory states are termed the *causal states* of the process, and the corresponding memory cost $D_\mu$ is termed the *topological complexity* of the process[9,10].

Renewal processes[11] represent a particularly apt class of stochastic processes for studying the impact of non-Markovianity in stochastic modelling. They generalise Poisson processes to time-dependent decay rates. In discrete-time, families of renewal processes with tunable lengths of memory effects can be constructed, providing a means of exploring how memory costs change as non-Markovianity is increased[4,12]. Renewal processes consist of a series of 'tick' events (labelled "1") stochastically spaced in time; in discrete-time, timesteps where no tick occurs are denoted "0". The time between each consecutive pair of events is drawn from the same distribution. Thus, a discrete-time renewal process is fully characterised by a *survival distribution* $\Phi(n)$, codifying the probability that two consecutive tick events are at least $n$ timesteps apart.

In this work, we consider a family of renewal processes with a periodically modulated decay (PMD) rate, which we refer to as PMD

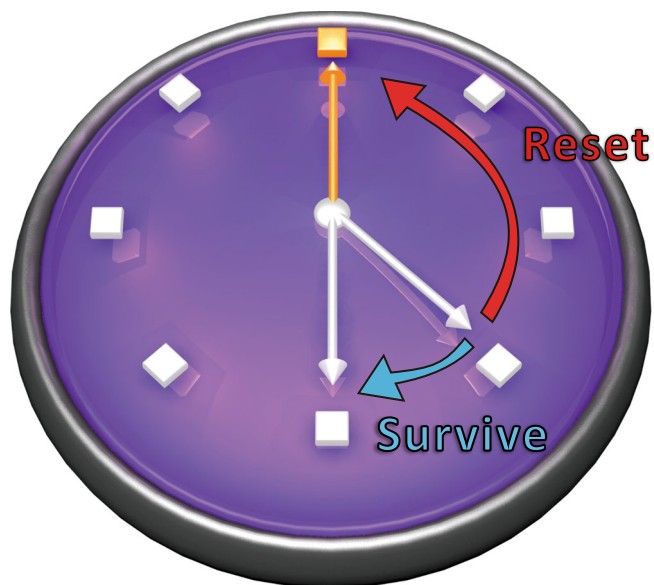

**Fig. 1 | Modelling PMD processes.** A PMD process with period $N$ can be exactly modelled with $N$ memory states. At each time step, the process either "survives" (no tick occurs) and the model advances to the next state or undergoes a tick event, and the model moves to a reset state. Due to the $N$-periodic nature of the conditional statistics, the model also returns to the reset state after surviving $N$ timesteps, leading to the clock-like structure of the model dynamics as depicted.

processes. Their survival probability takes the form

$$\Phi(n) = \Gamma^n (1 - V \sin^2(n\theta)), \tag{1}$$

where $\theta := \pi/N$. Here, $\Gamma$ represents the base decay factor (i.e., the probability that the process survives to the next timestep in the absence of modulation), $V$ the strength of the modulation, and $N \in \mathbb{N}$ the period length. Note that a physical PMD process must satisfy $\Phi(n) < \Phi(n-1) \forall n \in \mathbb{N}$. If we consider the process as a discretisation of a continuous-time process with base decay rate $\gamma$, then $\Gamma = \exp(-\gamma \Delta t)$, where $\Delta t$ is the size of the timestep.

For a general renewal process, the causal states are synonymous with the number of timesteps since a tick event last occurred[12,13], as the conditional distribution for the number of timesteps until the next tick is unique for each $n$. Further refinement is not necessary as the inter-tick time interval distributions are all conditionally independent. However, due to the symmetry of PMD processes, the conditional distribution repeats every $N$ steps, and so the causal states group according to the value of $n \mod N$ (see Fig. 1). Correspondingly, the minimal classical memory cost for statistically-exact modelling of a PMD process is $D_\mu = \log_2 N$. We remark that while $N$ thus suggests an effective memory length for the process, PMD processes nevertheless have an infinite Markov order (the number of timesteps that must be removed from the most recent past such that the remaining part is conditionally independent of the future, formally given by $\min_n |P(X_{0:\infty}|X_{-n:0}) = P(X_{0:\infty}|X_{-\infty:0}))$ for any $N \neq 1$. This requires that a model must, in general, retain information in memory across multiple timesteps about its initial preparation that cannot be extracted from output sequences of any length.

Quantum models can push memory costs below classical limits[1,2,14]. They operate by encoding relevant past information into a set of quantum memory states (i.e., $\{s_j\} \rightarrow \{|\sigma_j\rangle\}$). By coupling the quantum memory system with an ancilla probe (initialised in a 'blank' state $|0\rangle$) at each timestep, the output statistics can be imprinted onto the probe state. Specifically, in state-of-the-art quantum models[2], an interaction $U$ produces a superposition of possible outputs (encoded in the ancilla state) entangled with corresponding

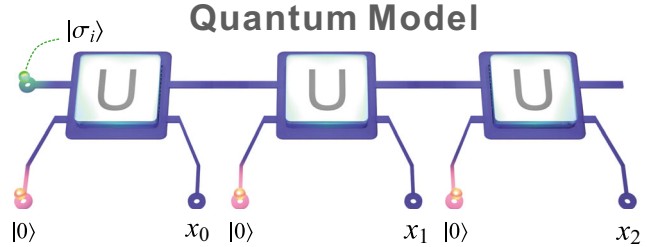

**Fig. 2 | Quantum models.** Quantum models store one of a set of memory states $\{|\sigma_j\rangle\}$ that correspond to an encoding of information from past observations (green). At each time step, a blank ancillary system set to $|0\rangle$ (red) is introduced and undergoes a joint interaction $U$ that creates a weighted superposition of possible events imprinted onto the ancilla, coupled with the corresponding updated memory state. The ancilla is then measured to produce the output statistics, leaving the memory ready for the next time step. Depicted are three timesteps, producing a string of outputs $x_0x_1x_2$.

updated memory states. Measurement of the ancilla (in the computational basis) then produces the output and leaves the memory system in the appropriately updated memory state. This procedure is repeated at each timestep using the same interaction and a fresh blank ancilla. See Fig. 2 for an illustration. Note that the interaction can equivalently be expressed in terms of Kraus operators $\{A_x := \langle x|U|0\rangle\}$ acting on the memory, where $\{x\}$ corresponds to the outputs. Further details, and the specific (tunable) form of $U$, can be found in the Methods section.

The form of $U$ implicitly defines (up to an irrelevant common unitary transformation) the quantum memory states $\{|\sigma_j\rangle\}$[2]. The memory cost of a quantum model is then given by the (logarithm of the) span of these states: $D_q = \log_2(\dim(\{|\sigma_j\rangle\}))$. Thus, when these quantum memory states are linearly dependent, $D_q$ is less than the corresponding classical cost[2–4]. We emphasise here the importance of linear dependence for quantum memory advantage: a quantum model will still require $2^{D_\mu}$ different memory states $\{|\sigma_j\rangle\}$ in one-to-one correspondence with the causal states, but when the quantum memory states are linearly dependent (such that they span a Hilbert space of dimension $2^{D_q} < 2^{D_\mu}$), a quantum memory advantage is achieved.

We show that PMD processes can be modelled with drastically reduced memory cost in this manner:

*Result (Theory)*: For any PMD process, we can construct a statistically-exact quantum model with memory cost $D_q \leq 1$.

That is, a statistically-exact quantum model can be constructed for any PMD process that requires only a single qubit memory. Crucially, this holds for any value of $N$, and so while the classical memory cost will diverge with increasing $N$, the quantum memory cost remains bounded. The quantum memory advantage $D_\mu - D_q$ is thus scalable.

We remark that this scalability comes with practical considerations. As $N$ increases, a quantum model using a single qubit as memory will necessarily require a high degree of overlap between some quantum memory states. This requires that an implementation of the model be able to store and manipulate quantum states with sufficiently high precision to meaningfully distinguish between these highly overlapping states, lest the impact of noise becomes too great. Thus, the theoretical scaling advantage is tempered by practical limitations on the precision afforded by their implementation. Nevertheless, as our ability to control quantum systems improves, we are able to ever increasingly offset these practical limits, and as our implementation shows, we can already begin mapping out the scaling curve.

For PMD processes with periodicity $N$, a quantum model can be specified by a pair of Kraus operators $\{A_0, A_1\}$ corresponding to each of the two outputs and a set of $N$ memory states $\{|\sigma_n\rangle\}$. Following the transition structure of the corresponding minimal-memory classical

model, these must satisfy

$$A_0|\sigma_n\rangle \propto |\sigma_{n+1 \bmod N}\rangle \tag{2}$$

$$A_1|\sigma_n\rangle \propto |\sigma_0\rangle. \tag{3}$$

That is, on event 0, the state label increments by 1 (modulo the periodicity), while on event 1, the state label resets to 0. In the Supplementary Material we show that for any PMD process—irrespective of the parameters—a set of such Kraus operators and quantum memory states exist within a 2-dimensional Hilbert space that will reproduce the correct output statistics for the process; in other words, a statistically exact quantum model with $D_q \leq 1$ can be constructed for any PMD process. Moreover, we provide an explicit construction of the Kraus operators and quantum memory states that we then use to design our implementation of the quantum models. This constitutes our main theory result.

### Experimental Implementation

We implement these memory-efficient quantum models of PMD processes using a quantum photonic setup. The experimental setup, illustrated in Fig. 3, consists of three modules: state preparation (orange), simulator (blue), and state tomography (green). The polarization of a photon is used for the memory qubit, and the ancilla(e) is encoded in its path degree of freedom.

The state preparation module is able to initialise the memory qubit in an arbitrary pure state, together with an initial vacuum state of the ancilla. This allows us to initialise the model in the state $|\sigma_j\rangle|0\rangle$ for any of the memory states $\{|\sigma_j\rangle\}$.

The simulation module is the key part of the model, where the photon undergoes an evolution to produce the outputs and updated memory state. At each timestep, the photon passes through a series of optical components that displaces the beam such that the path corresponds to the outputs $\{0, 1\}$, and the polarization is conditionally rotated into the subsequent memory state for the next time step. The details of this evolution are given in the Methods. Note that we do not measure the output ancilla until after the full simulation state, instead preserving the superposition over outputs. Thus, for an $L$-timestep simulation with the outputs mapped to path states, $2^L$ paths are needed in order to maintain this superposition. Nevertheless, it does not destroy the simulation if coherence is lost between the optical paths carrying different outputs, as the simulation does not require that they interact after their generation.

The final state tomography module enables us to validate the performance of the model. First, by detecting the final path of the photon, it manifests the output of the model, the statistics of which can then be checked. Second, through tomographic reconstruction of the final polarisation of the photon (conditional for each initial state and set of outputs), we are able to verify the integrity of the final memory state, which could, in principle, have instead been used to produce the outputs for further timesteps. We remark that as there are no nondeterministic elements to the evolution in our simulation stage, the impediments to running for larger $L$ are largely practical, in terms of the need for additional optical paths and optical equipment, and the accumulation of errors. We emphasise that tomography is used here only as a diagnostic of our experiment; in normal operation, measurement of the final path state of the photon alone is sufficient to extract the output of the model.

Our implementation runs the model for $L = 2$ timesteps. This is sufficient to witness the effect of memory preserved across timesteps; the conditional distribution of the second output given the first changes based on the initial memory state, indicating that information contained within this initial state is propagated across the simulation—i.e., that there is persistent memory. We modelled multiple PMD

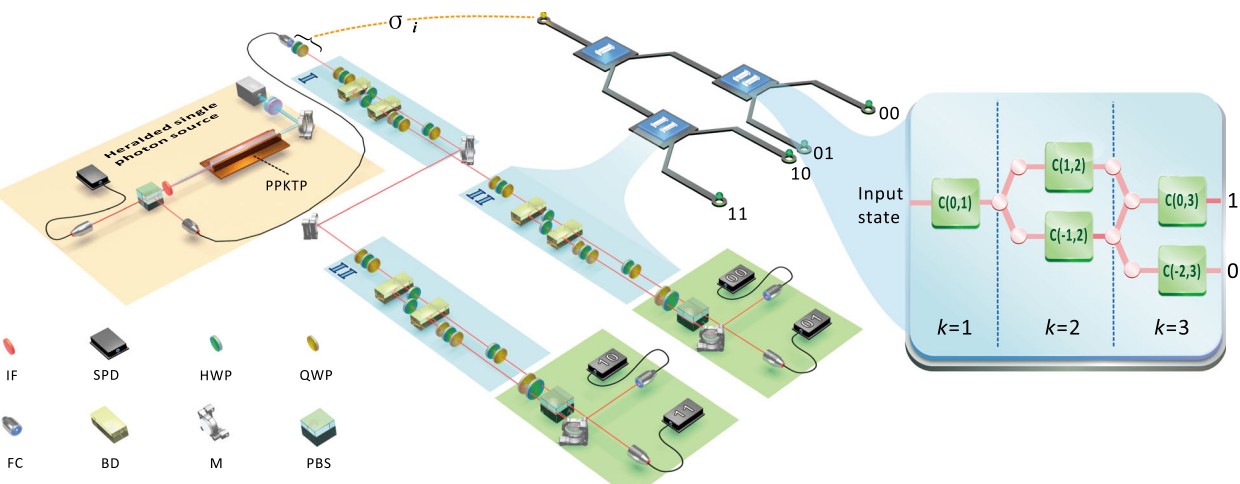

**Fig. 3 | Photonic implementation of quantum models of PMD processes.** We use a photonic setup to implement our quantum models. The orange region highlights the state preparation module, where two photons with a central wavelength of 808nm are generated via pumping a PPKTP crystal with temperature stabilised to around 35 °C through a type-II spontaneous parametric down-conversion process. One of the photons passes through a single-mode fibre and is prepared with an initial memory state encoded in its polarization, whilst the other is used as a trigger. The blue regions show the simulation module that carries out the evolution, encoding outputs into the photon path and updating the memory by rotating its polarisation (see Methods for details). After evolving two timesteps, the photon is passed into the tomography module (green region), where the output statistics are produced by photodetection counts, and the polarization is measured to tomographically reconstruct the final memory state. The optical components shown comprise PBS, polarising beamsplitter; M, mirror; IF, interference filter; QWP, quarter-wave plate; HWP, half-wave plate; FC, fibre coupler; BD, beam displacer; SPD, single photon detector.

processes with base decay factor Γ ranging from 0.49 to 0.64, period $N$ from 3 to 8, and modulation strength $V = 0.4$.

We briefly highlight key advancements our implementation makes over a prior experimental implementation of quantum dimension reduction[6]. While this prior work successfully demonstrated quantum dimension reduction in stochastic simulation – and made valuable experimental progress in doing so−it did not strictly make use of memory. That is, it simulated a Markovian process, and only for one timestep. Crucially, this did not require a persistent memory to be maintained across the evolution of a timestep (it is consumed and then reconstituted from the subsequent output), nor did it explicitly propagate a memory between multiple timesteps of an implementation. Thus, in this sense, only our implementation strictly demonstrates the memoryful simulation of a non-Markovian stochastic process with quantum dimension reduction. Further, it is only in our work that we demonstrate the superior accuracy of our implemented quantum models relative to that of the best classical models of the same dimension. We also remark on a secondary advantage specific to our implementation, namely that because we do not require any non-deterministic optical operations in our evolution, we avoid the exponential decay with $L$ of the probability of a successful simulation that would be suffered by this prior work had they sought to extend their simulation to further timesteps. This places our work much more favourably as a means to truly demonstrate the scalability of quantum dimension reduction.

**Experimental results**

We first verify that the output statistics produced by our model are faithful to the process. Outputs are determined by measurement of the final path of the photon, each corresponding to one of the four possible outputs for two timesteps of the process {00, 01, 10, 11}. For each of the parameter ranges detailed above and for each of the initial memory states $\{|\sigma_j\rangle\}$ we obtain $O(10^6)$ coincidence events, each corresponding to a single simulation run. We use these to reconstruct the probability distributions $\tilde{P}(x_0 x_1 | s_j)$. Figure 4a presents our obtained distributions for $N = 4$, $V = 0.4$ and Γ = {0.49, 0.52, 0.57, 0.64}, with the insets showing the discrepancy with the exact statistics. We quantify

this distortion of the statistics using the Kullbach–Liebler (KL) divergence[15] between experimentally-reconstructed and exact theoretical distributions (see Methods). We plot the normalised (per symbol) KL divergence $d_{KL}$ of our models in Fig. 5, where we see that for all parameters simulated, our models yielded a distortion below $10^{-2}$ bits.

Given this statistical distortion due to experimental imperfections, it would be disingenuous to consider only the memory cost of statistically exact classical models. In order to provide a fair comparison, we compare the accuracy we achieve to that of the least-distorted classical models with the same memory cost $D = 1$ (i.e., one bit). Specifically, we establish a lower bound on the smallest distortion (according to the KL divergence) that can be achieved by classical models with a single bit of memory (see Methods). This bound is plotted together with the distortion of our quantum models in Fig. 5, where we can see that our quantum models, in all cases, have a smaller distortion. That is, even accounting for the experimental imperfections of current quantum technologies, our quantum models of PMD processes achieve greater accuracy than is possible with any classical model of the same memory size. We remark that across all prior implementations of quantum models of stochastic processes, ours is the first to verify this. Note that the distortion in the classical models here is fundamental due to the constraints on the memory size, while for the quantum case, the distortion is purely due to imperfect experimental realisation.

We also verify the integrity of the final memory state at the end of our simulations. While we run our models for $L = 2$ timesteps, in principle, they can be run for arbitrarily many timesteps given sufficient optical components as the simulation updates the memory state at each step. This continuation requires that the final memory state output by the model (i.e., the polarisation of the photon) is faithful. By tomographic reconstruction of the photon polarisation, we can evaluate the infidelity of the final memory state $\tilde{\rho}$: $I(\tilde{\rho}) = 1 - \langle \sigma_k | \tilde{\rho} | \sigma_k \rangle$, where $|\sigma_k\rangle$ is the requisite final memory state given the initial state and outputs. In Fig. 4b, we plot the obtained infidelities for each initial state and outputs for $N = 4$, Γ = 0.49, and $V = 0.4$, while Fig. 4(c) shows the tomographically-reconstructed final memory state for each output when the initial state is $|\sigma_2\rangle$. We find that reconstructed final states are

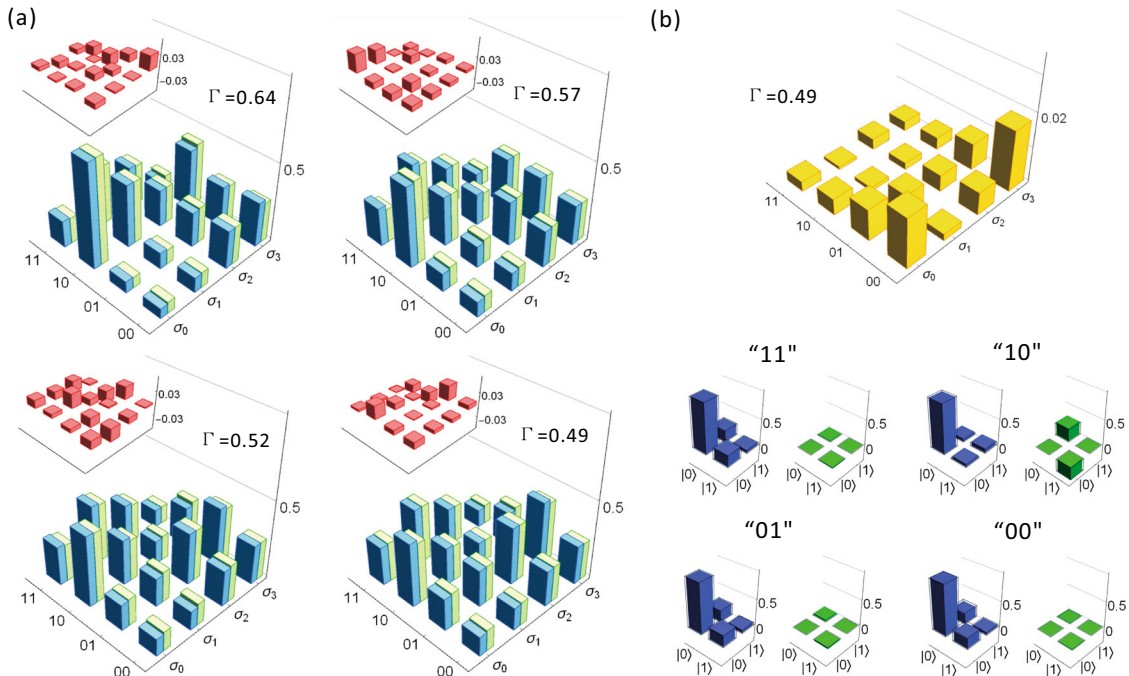

**Fig. 4 | Experimental results for quantum models of PMD processes.**
**a** Theoretical (green) and experimentally obtained (blue) probability distribution for two timestep simulations of PMD processes for each possible initial memory state. Insets show discrepancies between theoretical and experimentally-obtained values. Parameter range $N = 4$, $V = 0.4$, and $\Gamma = \{0.49, 0.52, 0.57, 0.64\}$. **b** Upper:

Infidelity of final memory states after two timesteps. Lower: Real and imaginary components of the tomographically-reconstructed final memory states after two timesteps for the initial state $|\sigma_2\rangle$; outlines show target values. Parameters: $N = 4$, $V = 0.4$ and $\Gamma = 0.49$.

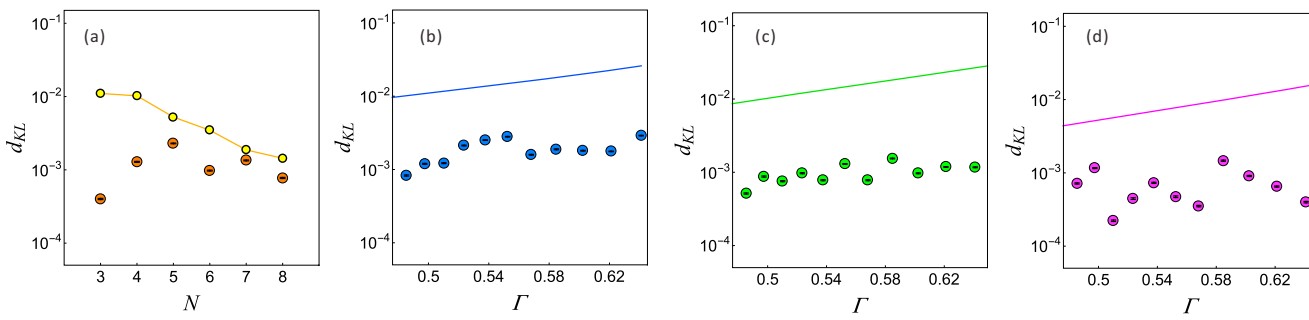

**Fig. 5 | Distortion of single (qu)bit memory models. a** KL divergence $d_{KL}$ of experimentally-obtained statistics from our quantum models (orange) from exact statistics, and lower bound on the divergence of single bit distorted classical models (yellow) for $N = [3..8]$, $\Gamma = 0.5$, and $V = 0.4$. **b–d** Analogous plots for $N = 3$

(b), $N = 4$ (c), and $N = 5$ (d) with varying $\Gamma$. Distortions of quantum models are shown as discs and lower bounds on the distortion of single-bit classical models as solid lines. Error bars are omitted as they are smaller than data points.

highly faithful to their corresponding requisite states (across all parameters simulated, a maximum infidelity of 0.0212 was obtained), suggesting that our simulation could be run for several more timesteps before the onset of significant degradation in the statistics.

## Discussion

Our work reports the first experimental implementation of quantum simulators of non-Markovian stochastic processes exhibiting memory advantages over optimal classical counterparts. We used these simulators to model a family of stochastic processes that have a tunable memory length; while increasing this corresponds to an ever-increasing classical memory cost, our simulators always require only a single qubit of memory—leading to a scalable quantum advantage. The non-Markovian nature of the processes required that information was retained in memory and propagated across the whole of the simulation over multiple timesteps that could not be extracted from

observing the outputs alone. Moreover, we show that this advantage is robust to the experimental noise introduced by our implementation via a comparison with bounds on the smallest noise achievable with classical models of the same memory cost.

The photonic setup in which we have implemented our quantum models is well-suited to the task at hand. As such models consist of repeated motifs of the interaction between memory and probe at each timestep—which are fixed in advance—the optical components can be finely calibrated in advance and achieve much smaller errors than typical of current universal quantum processors. Furthermore, our setup can readily be modified to simulate other non-Markovian stochastic processes. In particular, whilst not every renewal process can be exactly modelled by a quantum model with a single qubit of memory, recent work has developed techniques for constructing highly accurate *near*-exact quantum models of such processes with significantly reduced memory cost[5]. By adjusting only single-qubit

unitaries acting on photon polarisation—a comparatively straightforward task—our setup can implement single-qubit-memory quantum models (exact if possible, approximate otherwise) of *any* renewal process.

Our theoretical result on the scalability of the quantum advantage notwithstanding, there are still practical obstacles to a full experimental demonstration of the scaling. Namely, as $N$ increases, the proximate (in the label) quantum memory states have an increasingly strong overlap in statistics and, correspondingly, increasingly strong state overlap. Once the state (non-)overlap becomes comparable to the loss of fidelity in the evolution, the memory states will, in effect, 'smear' and lose the proper transition structure. While this will not be immediately clear in short output strings (as the statistics of the smeared states will look very similar), it will become increasingly apparent for larger $L$; thus, a proper test of the scalability of the advantage must also show the faithfulness of the statistics over longer numbers of timesteps. Increasing $L$ presents further challenges as the number of optical paths (and optical equipment) required grows exponentially with $L$. A preferable approach would be to fold the interactions into a recursive circuit that reuses the optical equipment at each timestep and avoids the exponential growth in the required number of optical paths. However, realising this by coupling the output into additional photons presents its own drawbacks[6], in terms of nondeterministic gates and the need to produce additional photons for each timestep.

A further advantage of our quantum models not explored here is that the outputs are not measured until the final step, up until which the output system is in a weighted superposition of the possible output strings[16]. This quantum sample (or '*q-sample*') state can then be used as an input to quantum algorithms for, e.g., quantum-enhanced analysis of the properties of the process[17] with potential applications in financial modelling[18,19]. Such q-samples (of length $L$) require a coherent superposition over all possible length $L$ output strings, which may present a challenge for current quantum hardware; nevertheless, we emphasise that the main task considered here—that of simulating the process' statistics—does not require this superposition, and requires only coherence in the memory qubit (i.e., photon polarisation) state.

Quantum models of stochastic processes have also been shown to exhibit other advantages over classical models that can be explored, such as reduced thermal dissipation[20,21]. We also note a close connection with studies on the fundamental limits of classical and quantum clocks[22–24], the latter of which have been shown to exhibit memory/accuracy advantages. Specifically, the behaviour of so-called 'reset clocks'—that are shown to be classically optimal[22] and postulated to be quantumly optimal—correspond to renewal processes; our models could be used to implement such reset quantum clocks.

Another enticing next step that builds upon our work is to extend to higher-dimensional quantum memories[25]. Further, by introducing a means of conditionally modifying the interaction, input-dependent stochastic processes can be implemented, which can be used to realise memory-efficient quantum-enhanced adaptive agents[26], complementing quantum techniques for accelerating their learning process[27,28]. Similar approaches could be made to implement simulators of *quantum* stochastic processes[29], which show interesting quirks in terms of non-Markovianity and Markov order[30,31]. Our work represents a key movement towards all these directions and applications.

## Methods

### Stochastic processes and minimal-memory classical modelling
A discrete-time stochastic process[7] consists of a sequence of random variables $X_t$, corresponding to events drawn from a set $\mathcal{X}$, and indexed by a timestep $t \in [t_{\min} .. t_{\max}]$. The process is defined by a joint distribution of these random variables across all timesteps $P(X_{t_{\min}:t_{\max}})$, where $X_{t_1:t_2} := X_{t_1}, X_{t_1+1}, \ldots X_{t_2-1}$ represents the contiguous (across timesteps) series of events between timesteps $t_1$

and $t_2$. We consider stochastic processes that are bi-infinite, such that $t_{\min} = -\infty$ and $t_{\max} = \infty$, and stationary (time-invariant), such that $P(X_{0:L}) = P(X_{t:t+L}) \, \forall \, t, L \in \mathbb{Z}$. Without loss of generality, we can take the present to be $t = 0$, such that the past is given by $\overleftarrow{x} := x_{-\infty:0}$, and the future $\overrightarrow{x} = x_{0:\infty}$. Note that we use upper case for random variables and lower case for the corresponding variates.

A (causal) model of such a (bi-infinite and stationary) discrete-time stochastic process consists of an encoding function $f : \overleftarrow{\mathcal{X}} \to \mathcal{S}$ that maps from the set of possible past observations $\overleftarrow{\mathcal{X}}$ to a set of memory states $s \in \mathcal{S}$[8–10]. The model also requires an update rule $\Lambda : \mathcal{S} \to \mathcal{S} \times \mathcal{X}$ that produces the outputs and updates the memory state accordingly. We then designate the memory cost $D_f$ of the encoding as the logarithm of the dimension (i.e., the number of (qu)bits) of the smallest system into which these memory states can be embedded[9]. For classical (i.e., mutually orthogonal) memory states, this corresponds to $D_f = |\mathcal{S}|$. For quantum memory states, which may, in general, be linearly dependent, $D_f \leq |\mathcal{S}|^2$.

Let us, for now, restrict our attention to statistically-exact models, such that $(f, \Lambda)$ must produce outputs with a distribution that is identical to the stochastic process being modelled. Under such a condition, the provably-memory minimal classical model of any given discrete-time stochastic process is known and can be systematically constructed[10]. These models are referred to as the *ε-machine* of the process, which employs an encoding function $f_\varepsilon$ based on the causal states of the process. This encoding function satisfies

$$f_\varepsilon(\overleftarrow{x}) = f_\varepsilon(\overleftarrow{x}') \iff P(\overrightarrow{X} | \overleftarrow{x}) = P(\overrightarrow{X} | \overleftarrow{x}'), \tag{4}$$

and given initial memory state $f_\varepsilon(\overleftarrow{x})$, the evolution produces output $x_0$ with probability $P(x_0 | \overleftarrow{x})$ and updates the memory to state $f_\varepsilon(\overleftarrow{x} x_0)$. The memory states are referred to as the causal states of the process, and the associated cost $D_\mu$ is given by the logarithm of the number of causal states.

### Classical models of PMD processes
Recall that a (discrete-time) renewal process is fully defined by its survival probability $\Phi(n)$, describing the probability that any consecutive pair of 1s in the output string is separated by at least $n$ 0s. We can deduce the distribution for the next output given the current number $n$ of 0s since the last 1:

$$P(0|n) = \frac{\Phi(n+1)}{\Phi(n)} \tag{5}$$

$$P(1|n) = 1 - \frac{\Phi(n+1)}{\Phi(n)}. \tag{6}$$

PMD processes correspond to a particular form of survival probability, viz., $\Phi(n) = \Gamma^n(1 - V\sin^2(n\theta))$. It can readily be seen that when inserted into Eq. (5), the output probabilities of PMD processes are periodic, with period $N$. Noting also that the counter $n$ always resets to 0 immediately after a 1 is output, we have that the causal state encoding function $f_\varepsilon$ maps pasts into memory states according to the value of $n \bmod N$, where $n$ is the number of 0s since the most recent 1. Without loss of generality, we can use this value to label the memory states $s_j$, with $j \in [0 .. N-1]$. Thus, upon output 0 the memory state will update from $s_j$ to $s_{(j+1)\bmod N}$, and on output 1 it will update to $s_0$ irrespective of the initial memory state. The probability of each output depends on the initial memory state.

### Quantum models
Though $\varepsilon$-machines are the provably memory-minimal classical models of stochastic processes, the memory cost can be pushed even lower through the use of quantum models—even when the process being

modelled is classical. Current state-of-the-art quantum models map causal states $\{s_j\}$ to corresponding quantum memory states $\{|\sigma_j\rangle\}$, which are stored in the memory of the quantum model[2]. The quantum model then functions by means of a unitary interaction between the memory system and an ancilla initialised in $|0\rangle$.

This unitary interaction takes the following form:

$$U|\sigma_j\rangle|0\rangle = \sum_x \sqrt{P(x|s_j)}e^{i\varphi_{xj}}|\sigma_{\lambda(j,x)}\rangle|x\rangle, \qquad (7)$$

where $\{\varphi_{xj}\}$ are a set of phase parameters that can be tuned to modify the memory cost, and $\lambda(j,x)$ is an update rule that returns the label of the updated memory state given initial state label $j$ and output $x$, following the transition structure of the corresponding $\varepsilon$-machine. As remarked above, Eq. (7) implicitly defines the explicit form of the quantum memory states $\{|\sigma_j\rangle\}$ up to an irrelevant common unitary transformation, as well as $U$. As the evolution always begins with the ancilla in the same blank state, it can also be equivalently be specified according to its Kraus operators $\{A_x := \langle x|U|0\rangle\}$, where the contractions are made only on the ancilla subsystem[32]. These Kraus operators satisfy the completeness relation $\sum_x A_x^\dagger A_x = \mathbb{1}$.

Following the definition of the memory cost of a model, the memory cost $D_q$ of such quantum models is given by the number of qubits required by a memory system to store the quantum memory states. In other words, the quantum memory cost is given by the logarithm of the number of dimensions in the smallest Hilbert space that can support the quantum memory states:

$$D_q = \log_2(\dim(\{|\sigma_j\rangle\})). \qquad (8)$$

This is upper-bounded by the memory cost of the $\varepsilon$-machine as $N$ quantum states span at most $N$ dimensions. That is, $D_q \leq D_\mu$, with equality if the quantum memory states are all linearly independent. For many stochastic processes, though, it is possible to find sets of linearly-dependent memory states satisfying Eq. (7), leading to a strict (and sometimes extreme) quantum memory advantage[2–4]—as we have also demonstrated for PMD processes.

### Further experimental details
The state preparation module of our implementation prepares the initial quantum memory state $|\sigma_j\rangle$ prior to the start of the simulation. We encode this memory state in the polarisation of a photon. To do this, a photon pair is prepared via spontaneous parametric down-conversion by pumping a type-II PPKTP crystal with a 404nm laser pulse. One of the two photons is first sent to a Glan-Thompson prism (GT) to ensure the photon is H-polarised; then, the H-polarised photon passes through a half-wave plate and a quarter-wave plate (H-Q). This allows us to prepare arbitrary initial qubit states in the photon polarisation[33]. Meanwhile, the other photon of the pair serves as a trigger and is detected via a single-photon detector (SPD).

As shown in the inset of Fig. 3, our implementation embeds the simulation module within a one-dimensional discrete-time quantum walk, using recently-introduced photonic techniques to realise arbitrary general evolution on one- and two-qubit systems[34–36]. Each timestep of the evolution is realised through a finite-step quantum walk evolution; the details of this embedding are given in the Supplementary Material.

### Quantifying statistical accuracy with KL divergence
We use the KL divergence to quantify the statistical accuracy of the output of a model (or realisation thereof). The KL divergence $\mathcal{D}_{KL}$ between a probability distribution $Q$ and a target distribution $P$ is given by[15]

$$\mathcal{D}_{KL}(P||Q) := \sum_x P(x)\log\left(\frac{P(x)}{Q(x)}\right). \qquad (9)$$

We must make two modifications to this to account for the fact that we deal with stochastic processes rather than straightforward distributions. First, we must apply it to conditional distributions based on the initial memory state (and subsequently average over the memory state distribution). Secondly, while a process constitutes an infinite string of outputs, we observe only a finite-length string. To account for this, we calculate the KL divergence over finite length strings and normalise to obtain a per-symbol divergence. Thus, we have

$$d_{KL}(P||\tilde{P};L) := \frac{1}{L}\sum_{s_j}\pi(s_j)\sum_{x_{0:L}}P(x_{0:L}|s_j)\log_2\left(\frac{P(x_{0:L}|s_j)}{\tilde{P}(x_{0:L}|s_j)}\right), \qquad (10)$$

where $\pi$ represents the steady-state distribution of the model's memory states. In our implementation, we run the simulation for two timesteps, and so we use $L = 2$. For general renewal processes without periodicity, the steady-state distribution $\pi(s_n) = \mu\Phi(n)$, where $\mu^{-1} := \sum_n \Phi(n)$ is a normalisation factor[12,13]. For PMD processes, this simplifies to $\pi(s_n) = \bar{\mu}\Phi(n)$, with $\bar{\mu}^{-1} := \sum_{n=0}^{N-1}\Phi(n)$.

### Classical models with distortion
As the implementations of our quantum models are subject to experimental noise—leading to distortion in the statistics—it is prudent to compare them against classical models with distortion. That is, rather than considering classical models of PMD processes with $N$ states that are able to produce statistically exact outputs, we consider imperfect classical models with only a single bit of memory available. This restriction on the memory unavoidably introduces distortion into the output statistics; we show that this distortion is greater than that of our implemented single-qubit-memory quantum models.

We use an approach akin to information bottleneck techniques[37] introduced in previous work[38] based on the concept of *pre-models* that are tasked with finding encodings of the past such that a string of future outputs (of pre-defined length $L$) can be produced from this encoded representation of the past. Such pre-models encompass models as a special case but are more general as they are not required to produce the outputs one timestep at a time, nor necessarily produce an arbitrarily-long string of future outputs. The minimal distortion of all $L$-step pre-models at the fixed memory cost of a given stochastic process serves as a lower bound on the smallest achievable distortion of a model with this memory cost.

The full details can be found in ref. 38, but intuitively, the mechanism of this approach can be understood as follows. We are seeking a combination of a map from the set of pasts $\{\overleftarrow{x}\}$ to a set of $\tilde{N} < N$ (for a bit, $\tilde{N} = 2$) memory states $\tilde{S}$ and an update rule $\tilde{\Lambda} : \tilde{S} \to \tilde{S} \times \mathcal{X}$ that produces the next output and updates the memory state, such that the error in the conditional distribution for the future outputs $\overrightarrow{X}$ given any particular past is minimised. Given the causal states are already a coarse-graining of the set of pasts into groups with statistically indistinguishable futures, we can constrain the initial map to assign any two pasts belonging to the same causal state $s$ to the same distorted memory state $\tilde{s}$. Consider now, that models producing the entire future one output at a time are a strict subset of models that produce the future in blocks of length $L$—which, as discussed above, are a special case of pre-models that produce only the next $L$-length block of outputs (all at once). Thus, the minimum distortion possible for an $L$-length pre-model lower the bound of the distortion of any model. This greatly simplifies the search space to need only consider mappings from $\mathcal{S} \to \tilde{\mathcal{S}}$, and instead of an update rule, only a map from $\tilde{\mathcal{S}}$ to the distribution of $L$-length futures $\mathcal{X}_{0:L}$.

With this approach, we are able to bound the distortion $d_{KL}^c$ achievable with single bit classical models of PMD processes. Formally, a $L$-step pre-model consists of an encoding function $\tilde{f} : \overline{\mathcal{X}} \to \mathcal{R}$, where $r \in \mathcal{R}$ are the memory states of the pre-model, and a set of conditional output distributions $\{Q_L(X_{0:L}|r)\}$. It has been formally proven that the minimum distortion (classical) pre-models have memory states that are a coarsening of the causal states. Thus, a lower bound on the distortion of single-bit-memory classical models is given by

$$d_{KL}^c \geq \min_{\tilde{f},\{Q_L\}} \frac{1}{L} \sum_{s_j} \pi(s_j) \sum_{x_{0:L}} P(x_{0:L}|s_j) \log_2\left(\frac{P(x_{0:L}|s_j)}{Q_L(x_{0:L}|\tilde{f}(s_j))}\right), \quad (11)$$

subject to the constraint that the encoding function maps to only two memory states. With the modest number of states and $L$ considered here, the minimisation is highly amenable to an exhaustive numerical search, which we perform to determine the lower bounds on classical distortion presented in the main text. We use $L = 2$ for parity with our implementations of quantum models.

## Data availability
Data displayed in the plots are made available as a supplemental file. Further details and explanations of the data are available from the authors upon reasonable request. Source data are provided in this paper.

## Code availability
The code used to determine bounds on classical models with distortion is available at https://github.com/Yangchengran/LearningQuantumStochasticModellingCode. Further details and explanations of the code are available from the authors upon reasonable request.

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

## Acknowledgements

This work was funded by the University of Manchester Dame Kathleen Ollerenshaw Fellowship, the National Research Foundation, Singapore, and Agency for Science, Technology and Research (A*STAR) under its QEP2.0 programme (NRF2021-QEP2-02-P06), the Imperial College Borland Fellowship in Mathematics, grant FQXi-RFP-1809 from the Foundational Questions Institute and Fetzer Franklin Fund (a donor-advised fund of the Silicon Valley Community Foundation), and the Singapore Ministry of Education Tier 1 grants RG146/20 and RG77/22. The work at the University of Science and Technology of China was supported by the National Natural Science Foundation of China (Grants Nos. 12134014, 61905234, 11974335, and 12104439), the Key Research Programme of Frontier Sciences, CAS (Grant No. QYZDYSSW-SLH003), USTC Research Funds of the Double First-Class Initiative (Grant No. YD2030002007) and the Fundamental Research Funds for the Central Universities.

## Author contributions

K.-D.W. and C.Y. contributed equally to this work. K.-D.W. and R.-D.H. performed the experimental work under the guidance of G.-Y.X., C.-F.L. and G.-C. G., C.Y. performed the theoretical work under the guidance of M.G. and T.J.E., K.-D.W., C.Y., M.G., G.-Y.X. and T.J.E. conceived the project, and K.-D.W., C.Y. and T.J.E. drafted the paper, with revisions from G.-Y.X. and M.G. All authors approved the final version of the paper.

## Competing interests

The authors declare no competing interests.
