## [Peer Review File · Nature Communications]

Implementing quantum dimensionality reduction for non-Markovian stochastic simulationREVIEWER COMMENTS

Reviewer #1 (Remarks to the Author):

The authors of the paper "Implementing quantum dimensionality reduction for non-Markovian stochastic simulation" consider to solve a classical stochastic non-Markovian evolution, namely a Periodically Modulated Decay (PMD) process. For the period of N , the minimum classical memory is given by $\log(N)$. The authors claim that solving the problem on a quantum simulator provides the advantage by reducing the memory size into a single qubit. The protocol has been experimentally implemented on a photonic setup with two steps of the evolution.

There are several issues with the claims in this paper which I list them below:

1. There is a fundamental problem for the claim of advantage of quantum simulation over classical computing in solving the proposed PMD process. In the quantum evolution, proposed by the authors, it seems that one has to perform quantum tomography on the output of the simulator. Since quantum tomography demands exponential number of operations the advantage of the algorithm is highly under question and its scalability seems to be impossible. Therefore, it is really not clear whether this algorithm provides any practical advantage.
2. I found the proof for the claim that one qubit is enough to track the memory in the section S1 in the supplementary material very unclear. In what sense a set of quantum states $|\sigma_0\rangle, |\sigma_1\rangle, \dots, |\sigma_N\rangle$ can be described by a single qubit. This has to be clarified.
3. The presentation of the paper is not reader friendly. I suggest that some part of the Methods, in particular for explaining the model and its solution, is moved to the main text. The main text should be self-sufficient for understanding the protocol and the Methods should only describe the details. In the current version, the main text alone is confusing and is certainly not self-sufficient.
4. Regarding the experimental part, for $L=2$ the output can always be described by 1 qubit. So, I am not sure if $L=2$ in the experimental setup is enough to demonstrate quantum advantage. Perhaps, the difficulty of tomography is a bottle neck which then brings up the question of practicality of the protocol, explained in my first comment.

Based on these issues, I cannot recommend this version of the paper for publication in Nature Communications. Nonetheless, I still remain open to revise my decision if the authors think that they can answer the above comments satisfactorily.

Reviewer #2 (Remarks to the Author):

In their manuscript 'Implementing quantum dimensionality reduction for non-Markovian stochastic simulation', the authors prove analytically that a family of renewal processes can be simulated exactly in quantum mechanics with a two-dimensional memory, constituting a sizable advantage over optimal classical models, which require an N -dimensional memory, where N is the periodicity of the renewal process. These theoretical findings are then demonstrated to be exploitable in practical scenarios. Concretely, the memory efficient simulation of renewal processes is implemented in a photonic setup and shown to recreate the correct statistics of the considered stochastic process up to very low distortions, using only a qubit of memory. For fair comparison, these experimental results are then compared to the best distortion a classical model using only one bit of memory can achieve, with the result that, for a wide variety of process parameters, even under this fairer comparison the quantum simulation of the process still outperforms all classical approximate simulations that use the same sized memory.

Overall, the manuscript is very well-written and - up to rather minor points (see below) - key assumptions and reasonings are easy to follow. The presented results are topical, on the one hand because renewal processes have seen a lot of interest recently, predominantly in the study of quantum clocks, and on the other hand because they contribute to the growing body of works that demonstrate advantages in the quantum simulation of stochastic processes that are not only theoretical, but experimentally observable.

These positive aspects notwithstanding, I am -- as of yet -- not convinced of the conceptual novelty of the results of the manuscript. In particular, I fail to see how they fundamentally differ from the work presented in Ref. [6]. If I am not mistaken, there, also, an experimentally realizable memory dimension reduction for the simulation of classical stochastic processes is demonstrated. In particular, Fig. 1b) of Ref. [6], depicting the overall conceptual idea of the quantum simulator, is equivalent to the corresponding Fig. 2 in the current manuscript. This conceptual similarity is shortly addressed in the current work where it is stated that the results of Ref. [6] '...were limited to models of Markovian processes'. However, I somewhat disagree with this distinction, since, basically, both the experimental setup of Ref. [6], as well as the current one simulate the behavior of the respective epsilon machine, i.e., the hidden Markov model of the stochastic process at hand. In this sense, one could argue that both works 'only' simulate Markovian processes; or the point could be made that the distinction between Markovian and non-Markovian is not of fundamental relevance here, since, in the end, it is about the dimension that is spanned by the causal states, a concept that pertains to the corresponding hidden Markov model anyways.

As a consequence, while very nice, and surely novel with respect to the type of processes that are modeled as well as their tunability, the results of the current manuscript appear more like an incremental amelioration of Ref. [6] than a 'first experimental implementation of quantum simulators of non-Markovian stochastic processes' that 'represent[s] a key movement' towards many technological implementations. This 'incremental flavor' of the results is reinforced by the fact that 'only' two-time probabilities are recorded, which, above all compared to the considered 'memory lengths' that go up to 8 seems like too little information to properly judge the performance of the setup. Naturally, I am not trying to claim that these experiments are not an achievement in their own right, but as it stands, they do not strike me as a substantial improvement of those performed in Ref. [6].

I would thus rather recommend publication of the manuscript in a more specialized journal than Nature Communications. I do understand though, that some of the authors of the current manuscript are also co-authors of Ref. [6], thus having a deeper understanding of the nuances and fundamental differences between the respective results than me, and I am happy to be convinced that the current work is sufficiently distinct and novel with respect to previous results in the literature to warrant publication in Nature Communications.

Besides this concern with respect to the suitability of the work for Nature Communications, there are a couple of minor points that should be addressed:

- The authors claim that their model has a Markov order N , where N is a model parameter that can be tuned. While I do not disagree that a change of N changes the memory properties of the process at hand, I do not think that the respective processes have Markov order N . I might be mistaken, but at any point in time, the conditional probability of the next outcome depends on how many zeros (modulo N) have previously been seen. In order to deduce this number, though, one would have to have access to the full past (at least until the last occurrence 1), which might be arbitrarily long. Put differently, deducing what causal state one is in can generally not be done on a finite available past, leading to infinite Markov order, at least according to Def. [15] in the manuscript. I might be misunderstanding the notion of Markov order put forward here, but if so, this point should be discussed in slightly more detail to alleviate potential confusions.

- On p. 3, as well as in the Discussion, it is highlighted as a strength of the experimental implementation that the output of the ancilla is only measured at the very end of the implementation. How scalable is this experimental procedure? Naively, it seems like for any moderate number of

timesteps, storage of the corresponding state until the end of the experiment would require a huge, well-isolated quantum memory. Is this feasible on current day technology/does the employed platform already enable such storage beyond two timesteps?

- On a similar note, I was wondering if the reported quantum advantage is scalable with N . If I see it correctly, increase of N would make the causal states more and more similar, requiring highly precise state preparation as well as implementation of unitaries. In the limit of large N , I would basically assume this experiment to put out white noise since experimental errors do not allow anymore to even closely ensure proper transition between quantum causal states. Does this trade-off between memory size reduction and requisite precision fundamentally limit the range of applicability of the method, or can it somehow be corrected for? Given the precision of the employed hardware, is it possible to gauge up to what value of N one obtains 'meaningful' statistics?

- In Fig. 5, is there an intuitive explanation for the behavior of the distortion of the single qubit memory models? They seem to be somewhat periodic/non-monotonous with respect to both N and γ . Is there an obvious experimental reason for why this is to be expected?

- In the Discussion, the potential application of the results to quantum clocks is alluded to. Since this topic has seen a lot of traction recently, I think it would be insightful and beneficial to a wide array of readers if the authors could elaborate in a little bit more detail on how they envision their results to potentially impact this field.

- It is mentioned in the Discussion that the employed experimental setup can 'implement single-qubit-memory quantum models of any renewal process'. I might be mistaken, but as far as I can see, there is only a proof of this statement for the renewal process of Eq. (1), but not for all conceivable renewal processes.

- In the Methods, the steady-state distribution of the memory state is used. This distribution should be derived explicitly somewhere for the process at hand.

- The comparison to classical single-bit-memory models with distortion is very nice and provides a fair comparison between quantum and classical memory requirements. However, I think the discussion of how the bounds for classical models with distortion are obtained (p. 7, last two paragraphs) could benefit from a more detailed explanation. As it stands, there is not enough information provided in the manuscript to allow for an intuitive understanding as to why it is actually possible to bound all distorted classical models in a numerically tractable way. To be clear, I am not asking for a full derivation, just some more high-level explanation to create a well-rounded picture.

- Why can the eigenvalues of A_0 be chosen to be 1 and $\exp(i\phi)$ below Eq. (S2) without losing generality? Also, wouldn't this choice then most likely make η complex, while it is assumed to be real throughout the derivation?

- While a somewhat trivial step, I think it should be mentioned explicitly at the end of S1 that the two CP maps A_0 and A_1 can indeed always be implemented by means of a unitary that acts on the causal state and a qubit, but do not require a larger environment.

- In the second sentence of p. 2, there is a period missing between 'observations' and 'A'.

Response to Reviewer 1

Reviewer 1 writes:

“The authors of the paper *Implementing quantum dimensionality reduction for non-Markovian stochastic simulation* consider to solve a classical stochastic non-Markovian evolution, namely a Periodically Modulated Decay (PMD) process. For the period of N , the minimum classical memory is given by $\log(N)$. The authors claim that solving the problem on a quantum simulator provides the advantage by reducing the memory size into a single qubit. The protocol has been experimentally implemented on a photonic setup with two steps of the evolution.”

We thank the reviewer for their feedback in their report on our work. Their summary is a largely accurate overview of our work. For clarity, we remark that ‘model’ or ‘simulate’ would be a more faithful replacement of ‘solve’, but otherwise what is written is correct. Specifically, the task at hand is to design models of non-Markovian stochastic processes that can faithfully replicate their output statistics whilst requiring the minimal amount of memory to do so. We also note a further important result in our work, that we show the error due to our imperfect experimental implementation is still less than what is achievable with the best classical models that use only a single bit of memory; this demonstrates a robustness of idealised theoretical results to experimental practicalities.

We now address their specific comments on the manuscript below.

“1. There is a fundamental problem for the claim of advantage of quantum simulation over classical computing in solving the proposed PMD process. In the quantum evolution, proposed by the authors, it seems that one has to perform quantum tomography on the output of the simulator. Since quantum tomography demands exponential number of operations the advantage of the algorithm is highly under question and its scalability seems to be impossible. Therefore, it is really not clear whether this algorithm provides any practical advantage.”

We would like to make sure we are very clear on this point – the simulation of the process’ statistics, and the associated memory advantage **do not require quantum tomography**. The statistics are generated simply by measuring which path the photon is in at the end of the evolution. Quantum tomography is used here solely as a diagnostic tool of our experiment.

Let us elaborate further on this. The quantum models, as depicted in Fig. 2, consist at each timestep of a joint interaction between the memory system and a ‘blank’ ancilla set to $|0\rangle$, followed by a measurement of the ancilla system in a fixed basis – the output for the timestep is encoded into the ancilla state, and is manifest through this measurement. The memory system is then preserved and passed through into the next timestep, where it undergoes the same interaction with a fresh ancilla; Fig. 2 depicts the consecutive operation of 3 such timesteps. In our experiment, the polarisation of a photon is used as the memory system, and the photon path the ancillary system. The measurement made is that of which path the photon is in. The ancillary systems need not be measured immediately after the evolution; in our experiment we make a single measurement on the full ancillary space after the two evolution timesteps.

Thus, in our experiment with $L = 2$ timesteps, the photon is initially in a single path state, which branches into two paths after the first timestep, each of which then branch in two (for a total of four paths) after the second timestep. These four paths then correspond to the four possible outputs after two timesteps (00, 01, 10, and 11, where the first digit corresponds to the output for the first timestep, and similarly, the second digit for the second timestep). Measurement of the final path of the photon then manifests the output. This is all that is needed to produce the output statistics of the model – i.e., the task that is set of the quantum model.

Tomography then, only comes in as a diagnostic tool of the correct function of our experimental implementation of the model, and not as part of ‘normal’ operation of the model. To understand the purpose of this, consider that the model derived in the theory can in principle be implemented for an arbitrarily long number of timesteps. While our experimental realisation of the model only implements

two timesteps, we could in principle add additional optical equipment and run it for more timesteps (we remark that our revised manuscript now includes further discussion of the challenges involved in increasing L , in response to the comments of the second reviewer). To implement further steps with high accuracy, we would require that the state of the memory (encoded in the photon polarisation) at the end of the evolution for the prior timesteps is sufficiently accurate to ensure the additional outputs will also be statistically faithful. We use tomography to verify this. That is, after the $L = 2$ timesteps implemented, we are determining the fidelity between the actual final state of the memory (conditioned on the measured output and initial state) obtained in our experiment, and the theoretical ideal. Across all parameters implemented and all combinations of initial states and outputs we observe fidelities > 0.978 after two timesteps. This tomography is performed only on a single qubit (the photon polarisation) irrespective of L , N , and other parameters. We emphasise again that this tomography is purely a diagnostic tool for our implementation, and not required for the normal operation of the model (i.e., simulating the process' statistics), which requires only a measurement of the final photon path.

We have amended the manuscript to more clearly emphasise this.

“2. I found the proof for the claim that one qubit is enough to track the memory in the section S1 in the supplementary material very unclear. In what sense a set of quantum states $|\sigma_0\rangle, |\sigma_1\rangle, \dots, |\sigma_N\rangle$ can be described by a single qubit. This has to be clarified.”

In a sense, this comes down to the crux of how the quantum advantage works. Indeed, each of the N classical memory states are mapped to a different quantum memory state $|\sigma_n\rangle$. However, unlike the classical states, which are all mutually orthogonal, the corresponding quantum memory states are *highly* linearly dependent. Collectively, they span only a 2-dimensional Hilbert space, such that we can consistently express all of the quantum memory states in the form $|\sigma_n\rangle = \alpha_n|0\rangle + \beta_n|1\rangle$. Because of this, only a single qubit is needed to encode the quantum memory states. As an example, consider that $\{|0\rangle, |1\rangle, |+\rangle, |-\rangle\}$ are 4 different states, but all exist within the same 2-dimensional Hilbert space. In S1, the linear dependence of the memory states is implicitly built into the form of the Kraus operators, which can be expressed as 2×2 matrices. We have now further emphasised this point in the manuscript.

Crucially, note that the model is not required to extract the label n of the memory states from the $|\sigma_n\rangle$, only an effective bias that it provides to the output statistics. Indeed, the quantum advantage essentially relies on the fact that knowing n is generically more information than is necessary for producing the statistics. Specifically, each n requires a different memory state in order to differently bias the output statistics; classically, this cannot be compressed further than to have mutually orthogonal states for each n , while quantumly we are able to embed this bias within different states of a qubit.

“3. The presentation of the paper is not reader friendly. I suggest that some part of the Methods, in particular for explaining the model and its solution, is moved to the main text. The main text should be self-sufficient for understanding the protocol and the Methods should only describe the details. In the current version, the main text alone is confusing and is certainly not self-sufficient.”

We appreciate the reviewer's candour here. Our original presentation had sought to emphasise the results in the Results section, whilst leaving the technical theoretical details to the Methods. However, as it appears that this has come at the detriment of the readability of the work, we have moved some details from the Methods into the Framework and Theory subsection of the Results. The Results section now includes additional details that provide an intuitive picture of how the quantum models operate generally, and outline the details of how quantum models for PMD processes specifically are constructed. We believe this restructuring will now enable non-specialist readers to better appreciate the mechanics of the quantum models from the Results alone – without needing to delve into the full formal details underpinning them – while these further details can still be found in the Methods and Supplementary Material.

“4. Regarding the experimental part, for $L = 2$ the output can always be described by 1 qubit. So, I am not sure if $L = 2$ in the experimental setup is enough to demonstrate quantum advantage. Perhaps, the difficulty of tomography is a bottle neck which then

brings up the question of practicality of the protocol, explained in my first comment.”

$L = 2$ corresponds to two timesteps, with four possible outcomes ($\{00, 01, 10, 11\}$). Therefore, the output for $L = 2$ requires 2 (qu)bits to express. This is captured by the four possible paths the photon can be in at the end of the evolution. The qubit encoded within the polarisation of the photon constitutes the memory system of the model; as noted above, tomography of this over several runs is used only as a diagnostic test in our experiment, to check that this memory state is sufficiently close to what would be required to meaningfully produce statistics in additional timesteps, which would only require additional optical equipment arranged as per the blue blocks of Fig. 3. Moreover, this diagnostic tomography is performed only on the memory qubit (i.e., photon polarisation), not the full polarisation + path state. We believe however that $L = 2$ timesteps is sufficient to capture the essence of the experiment – i.e, demonstrating quantum dimensionality reduction in the simulation of non-Markovian stochastic processes.

“Based on these issues, I cannot recommend this version of the paper for publication in Nature Communications. Nonetheless, I still remain open to revise my decision if the authors think that they can answer the above comments satisfactorily.”

We again thank the reviewer for their report. We hope that our above response provides convincing answers to their comments, and thank them for stimulating the associated improvements to our manuscript.

Response to Reviewer 2

Reviewer 2 writes:

“In their manuscript ‘Implementing quantum dimensionality reduction for non-Markovian stochastic simulation’, the authors prove analytically that a family of renewal processes can be simulated exactly in quantum mechanics with a two-dimensional memory, constituting a sizable advantage over optimal classical models, which require an N -dimensional memory, where N is the periodicity of the renewal process. These theoretical findings are then demonstrated to be exploitable in practical scenarios. Concretely, the memory efficient simulation of renewal processes is implemented in a photonic setup and shown to recreate the correct statistics of the considered stochastic process up to very low distortions, using only a qubit of memory. For fair comparison, these experimental results are then compared to the best distortion a classical model using only one bit of memory can achieve, with the result that, for a wide variety of process parameters, even under this fairer comparison the quantum simulation of the process still outperforms all classical approximate simulations that use the same sized memory.

Overall, the manuscript is very well-written and - up to rather minor points (see below) - key assumptions and reasonings are easy to follow. The presented results are topical, on the one hand because renewal processes have seen a lot of interest recently, predominantly in the study of quantum clocks, and on the other hand because they contribute to the growing body of works that demonstrate advantages in the quantum simulation of stochastic processes that are not only theoretical, but experimentally observable.”

We thank the reviewer for their detailed feedback on our work in their report. Their summary of our work is an accurate overview. We are pleased by their positive remarks on our writing and presentation, and further, that they agree the results are of topical significance.

“These positive aspects notwithstanding, I am – as of yet – not convinced of the conceptual novelty of the results of the manuscript. In particular, I fail to see how they fundamentally differ from the work presented in Ref. [6]. If I am not mistaken, there, also, an experimentally realizable memory dimension reduction for the simulation of classical stochastic processes is demonstrated. In particular, Fig. 1b) of Ref. [6], depicting the overall conceptual idea of the quantum simulator, is equivalent to the corresponding Fig. 2 in the current manuscript. This conceptual similarity is shortly addressed in the current work where it is stated that the results of Ref. [6] ‘... were limited to models of Markovian processes’. However, I somewhat disagree with this distinction, since, basically, both the experimental setup of Ref. [6], as well as the current one simulate the behavior of the respective epsilon machine, i.e., the hidden Markov model of the stochastic process at hand. In this sense, one could argue that both works ‘only’ simulate Markovian processes; or the point could be made that the distinction between Markovian and non-Markovian is not of fundamental relevance here, since, in the end, it is about the dimension that is spanned by the causal states, a concept that pertains to the corresponding hidden Markov model anyways.

As a consequence, while very nice, and surely novel with respect to the type of processes that are modeled as well as their tunability, the results of the current manuscript appear more like an incremental amelioration of Ref. [6] than a ‘first experimental implementation of quantum simulators of non-Markovian stochastic processes’ that ‘represent[s] a key movement’ towards many technological implementations. This ‘incremental flavor’ of the results is reinforced by the fact that ‘only’ two-time probabilities are recorded, which, above all compared to the considered ‘memory lengths’ that go up to 8 seems like too little information to properly judge the performance of the setup. Naturally, I am not trying to claim that these experiments are not an achievement in their own right, but as it stands, they do not strike me as a substantial improvement of those performed in Ref. [6].

I would thus rather recommend publication of the manuscript in a more specialized journal than Nature Communications. I do understand though, that some of the authors of the

current manuscript are also co-authors of Ref. [6], thus having a deeper understanding of the nuances and fundamental differences between the respective results than me, and I am happy to be convinced that the current work is sufficiently distinct and novel with respect to previous results in the literature to warrant publication in Nature Communications.”

While we do not wish to downplay the achievements of Ref. [6] – it was indeed also a key step in the development of quantum stochastic models and their applications – the present work demonstrates a number of advancements over this earlier experiment. We relay these below, with the key points emphasised in bold; we hope these provide as sufficiently convincing for the reviewer as they do for us that the manuscript warrants publication in Nature Communications. We have also made amendments to the manuscript to better emphasise these advancements.

Firstly, Ref. [6] covers only a single timestep, and so the ‘memory’ is never actually used to propagate information between timesteps. In contrast, by simulating two consecutive timesteps, **only our experiment actually uses a memory to propagate information between timesteps**. While [6] does show that the relevant information for their process can be propagated using only a qubit (and is encoded as such at the start and end of their experiment), the retention of information between timesteps in the memory is not strictly demonstrated.

Moreover, the distinction between Markovian and non-Markovian is important. Figure 1b of [6]/Fig. 2 are indeed depicting the same thing, insofar as the quantum models consist of a closed (i.e., unitary) interaction between a memory system and probe; this is the general building block of such quantum models, and can be found in most recent theoretical works in the field, predating also Ref. [6]. But the specifics of this unitary *are* important, and there are key structural differences between the U of [6] and that of our manuscript. Indeed, [6] makes specific use of simplifications that can be made only for Markovian processes. For a Markovian process, the memory system does not need to be kept intact across the whole simulation, and can be reconstituted at each timestep solely from the output of that timestep, as this contains all relevant information about the past. In contrast, outputs from previous timesteps are also relevant for non-Markovian processes, and so the memory must retain information about these across multiple timesteps; it cannot be determined from the most recent timestep alone. **Ref. [6] explicitly reconstitutes the memory at the end of the timestep from the output system; this simplification cannot be done with non-Markovian processes such as those implemented in our work.** That is, Ref. [6] consists of a preparation of the initial memory state, which then is evolved into the output for the single timestep from this prepared state, followed by using this output to create the final memory state. On the other hand, for our experiment the final memory state cannot be determined from the outputs alone, and is contingent also on the initial memory state – requiring that information is preserved in memory across the full evolution. In other words, **only our experiment requires memory to be carried across the entirety of a/multiple timestep(s).**

On a more pedantic point related to the above, quantum models simulate the ε -machine, insofar as they produce statistically-identical outputs. But it is problematic to use this to equate hidden Markov with Markov; ultimately, every process (Markov or not) can be simulated by its ε -machine, which is evolved by a joint Markovian evolution on the memory and output space – we argue that it would be disingenuous to then say that this makes the distinction between Markovian and non-Markovian processes of no relevance for their simulation. From a more strict emulation perspective, the internal workings of quantum models are distinct from ε -machines. The quantum memory states – though in one-to-one correspondence with the causal states (and classical memory states) – are *not* the causal states, and the corresponding causal state fundamentally cannot be extracted from a quantum memory state.

Further, while both experiments are based on photonic technologies, there are key differences in how they are used to implement quantum models. For both setups, the memory is stored within the polarisation of a photon, and the outputs in the photon path state. However, while we use the same photon for the memory for the whole of each instance of the experiment, Ref. [6] introduces a new photon to become the new memory at each timestep. This means that in our setup, we only need two photons (the second used as a herald) in each experimental instance, and the only nondeterministic element is the generation of this photon pair – irrespective of how many timesteps we simulate. In contrast,

Ref. [6] uses 4 photons to implement a single timestep, together with a nondeterministic evolution that has a probability of failure; further timesteps of evolution would need additional photons and additional nondeterministic evolution. On the face of it, it might then appear that we simply trade off needing additional photons for more timesteps to instead require an ever growing number of optical paths. Yet while scaling the number of optical paths presents a practical difficulty (see discussion below), increasing the number of photons presents a fundamental difficulty in scaling. While the number of photons required would grow linearly with L , the probability of simultaneously generating them falls off exponentially. Similarly, the probability of all nondeterministic gates succeeding falls off exponentially with L . Thus, **the approach of Ref. [6] requires a nondeterministic evolution with success probability that decays exponentially with L (and is exponentially hard to initialise the requisite number of photons); our approach avoids this.**

As a more minor point, we also remark that the process implemented in our manuscript is (at least from a theoretical standpoint) suitable for demonstrating scalable advantage, unlike that of Ref. [6]. While we do not demonstrate this scaling advantage experimentally, we have made crucial steps towards this (to quote our Introduction, “*This [...] presents a key step towards demonstrating the scalability [...] of such quantum memory advantages*”), by implementing quantum models of a process that (theoretically) possesses such a scaling curve at the first few parameter points of this curve, and making the requisite advances detailed above. While further advances are yet needed to experimentally demonstrate scaling advantages, we have made solid progress towards this. It should also be noted that Ref. [6] specifically cites demonstration of the scaling advantage as a highlight for future work.

Finally, we also note that **only our manuscript demonstrates that our implemented quantum models are more accurate than fundamentally achievable with any classical model of the same memory dimension.** That is, we have shown that even with experimental imperfections, we still demonstrably have a quantum advantage, as we realise a more accurate simulation than possible using any classical model with equivalent memory constraints. While we are confident that the experiment of Ref. [6] would also have achieved this for much of the parameter range of the process simulated therein, it is not explicitly demonstrated. Thus, only our manuscript is able to establish this formally by deriving provable bounds on the accuracy classical models of the same dimension can achieve, and demonstrating that our experimentally-implemented quantum models surpass this bound.

The reviewer also comments that we ‘only’ simulate two timesteps ($L = 2$), in contrast to the $N = [3..8]$ effective memory length of the process. But the two are distinct concepts, and we do not believe the two should be considered synonymously. N can be taken as a proxy for the classical complexity (i.e., memory cost) of modelling the process – this we consider to be its significance in the context of our work. On the other hand, L represents the length of the implementation. Since the process has infinite Markov order (see response to comment below), the final memory state cannot be determined with certainty from the outputs alone, so there is nothing special about $L \geq N$, other than that the model may have cycled through all the quantum memory states (which occurs if there is a block of L consecutive 0s). Rather, the main feature we set out to demonstrate was the quantum memory advantage in the transmission of information that necessarily had to be passed through the memory system (as opposed to extractable from outputs) across and between multiple timesteps. This is what our experiment indeed achieves, and $L = 2$ is sufficient for this purpose. From this perspective, extending to $L = 3$ would be somewhat incremental to what we have achieved with $L = 2$, as it would not highlight new elements in the implementation of quantum models of stochastic processes. Nevertheless, by verifying the high integrity (fidelity $\gtrsim 0.98$) of the final memory state, we have shown that our memory system can in principle be subsequently used to implement larger L . Considering also the points raised in the above discussion, we therefore believe our $L = 2$ simulation of a non-Markovian process to be a significant advancement over (and not ‘incremental’ to) the $L = 1$ simulation of a Markovian process in Ref. [6]. We hope that we have now also convinced the reviewer of this.

We now address the further comments of the reviewer.

“The authors claim that their model has a Markov order N , where N is a model parameter that can be tuned. While I do not disagree that a change of N changes the memory properties of the process at hand, I do not think that the respective processes have Markov

order N . I might be mistaken, but at any point in time, the conditional probability of the next outcome depends on how many zeros (modulo N) have previously been seen. In order to deduce this number, though, one would have to have access to the full past (at least until the last occurrence 1), which might be arbitrarily long. Put differently, deducing what causal state one is in can generally not be done on a finite available past, leading to infinite Markov order, at least according to Def. [15] in the manuscript. I might be misunderstanding the notion of Markov order put forward here, but if so, this point should be discussed in slightly more detail to alleviate potential confusions.”

The reviewer is entirely correct here – the process is indeed generically infinite Markov order (the exception being if $\Phi(n)$ becomes exactly zero at some finite n), irrespective of N . This was a daft slip-up that the last author takes responsibility for introducing during the writing of the manuscript, and is now fixed. We thank the reviewer for spotting this. For avoidance of doubt, this does not impact any of the results of the manuscript; the relevant quantity is the classical memory cost, which remains $\log_2 N$ regardless.

“On p. 3, as well as in the Discussion, it is highlighted as a strength of the experimental implementation that the output of the ancilla is only measured at the very end of the implementation. How scalable is this experimental procedure? Naively, it seems like for any moderate number of timesteps, storage of the corresponding state until the end of the experiment would require a huge, well-isolated quantum memory. Is this feasible on current day technology/does the employed platform already enable such storage beyond two timesteps?”

Indeed, we consider that (at least in the ideal setting) the output is preserved in a superposition of all possible output strings to be an advantage, as the corresponding ‘q-sample’ state can in principle be used as a resource for quantum-enhanced analysis of the statistics (see e.g., Ref. [21]). But to be clear, applications using such q-samples are beyond the direct scope of this work, and are not integral to the main task at hand here, i.e., that of simulating the statistics. Nevertheless, it is of course preferable if our implementation can preserve the q-sample state as a precursor to these future applications.

In this vein, let us consider what actually scales with increasing L . Firstly, the memory system will always consist of a single qubit irrespective of L , encoded within the polarisation of a photon. On the other hand, since the number of outputs grows with L , so too must the size of a q-sample – specifically, it requires L qubits [note that this scaling is a property of the q-sample, not specific to the implementation]. These outputs are encoded within the photon path, and so the number of paths does increase with L . The number of optical components required for the evolution will also scale with the number of timesteps.

So, generation of an L -step q-sample requires us to maintain coherence over and $L + 1$ qubit state (L qubits in the photon path, 1 in the polarisation). Yet, for the primary task of simulating the statistics, the coherence of the output state is irrelevant, and while losing coherence between the photon paths will destroy the q-sample, it will not affect the output statistics. All that needs to be maintained is the coherence in the memory state (i.e., the polarisation of a single photon), which is a much more amenable requirement.

In principle then, our experiment can be readily scaled to larger L with additional photon paths and additional optical equipment. Our present setup uses a comparatively modest amount of optical equipment – considerably less than e.g., state-of-the-art boson sampling experiments – and a few more timesteps can be added before we reach the same multitudes of paths and beamsplitters found in such experiments. Nevertheless, this does indeed eventually limit the practical scaling in how many timesteps can be implemented. A stopgap solution would be to also use time-delay loops to artificially increase the number of available paths, but this also would not be indefinitely scalable. Ultimately, the longer-term path to scalability in producing the output statistics is to look to recurrent circuits where the output is measured after each timestep, as depicted in the theoretical model of Fig. 2 – and this is a consideration for future experiments in the field. From the perspective of q-sample generation though, the present scaling of the output system size is unavoidable; the L -step of a process with alphabet \mathcal{A} will fundamentally need $|\mathcal{A}|^L$ states irrespective of how it is generated.

We remark that the choice of $L = 2$ was not purely pragmatic – we felt that the essence of the experimental goals were already achieved with $L = 2$, with larger L yielding diminishing returns in terms of demonstrating further scientific value beyond this.

We have incorporated points from this discussion within the manuscript.

“On a similar note, I was wondering if the reported quantum advantage is scalable with N . If I see it correctly, increase of N would make the causal states more and more similar, requiring highly precise state preparation as well as implementation of unitaries. In the limit of large N , I would basically assume this experiment to put out white noise since experimental errors do not allow anymore to even closely ensure proper transition between quantum causal states. Does this trade-off between memory size reduction and requisite precision fundamentally limit the range of applicability of the method, or can it somehow be corrected for? Given the precision of the employed hardware, is it possible to gauge up to what value of N one obtains ‘meaningful’ statistics?”

Yes, larger N will result in a greater overlap in the statistics of nearby (in terms of label n) causal states, with the corresponding quantum memory states similarly having high overlap. Eventually, as N is increased we will reach the point where the nearby quantum memory states have an overlap greater than the fidelity to which we can implement the evolution, indeed putting a practical limit on the scaling with N . But this is a practical limit, not a fundamental limit, and can be mitigated by a refined experimental setup, which could include e.g., higher-precision optics, error correction to protect the memory state, and gate error mitigation techniques such as zero noise extrapolation. Of course, such measures will bear additional resource requirements, such as a higher memory cost that would lead to a further accuracy/memory trade-off. We very much view research in these directions as highly valuable, and they will be a future focus.

With the present setup, a rigorous analysis of the point at which we can no longer obtain meaningful statistics would require a much more comprehensive model of the error in the evolution than we have to hand. But we can make a very naïve ‘back-of-the-envelope’ estimate based on what we have to hand in the results. We have that the final memory state experimentally reconstructed after two timesteps has a fidelity $\gtrsim 0.98$ with the theoretical ideal. Let us assume that the N quantum memory states are evenly spaced around a great circle of the Bloch sphere [an oversimplification, but likely balances out with the observation that the causal states of a renewal process with greatest steady-state probability mass are typically those which have smallest statistical overlap]. In this naïve picture, neighbouring memory states all have an infidelity $\sin^2(\pi/N) \approx \pi^2/N^2$. This suggests that once we reach $N \sim 20$, the loss of fidelity due to the evolution will be comparable to the distance between memory states, and so they will begin to smear together. However, it should also be noted that high overlap of memory states implies high overlap in statistics – and so this smearing will not be very visible in the output statistics until a larger number of timesteps are implemented.

We have incorporated points from this discussion within the manuscript.

“In Fig. 5, is there an intuitive explanation for the behavior of the distortion of the single qubit memory models? They seem to be somewhat periodic/non-monotonous with respect to both N and Γ . Is there an obvious experimental reason for why this is to be expected?”

Sadly, we do not have a definitive explanation for the behaviour of the distortion of the quantum models with respect to N and Γ . It should be noted though that unlike the single bit classical models, the distortion here is not fundamental, in the sense that a perfectly implemented quantum evolution would show no distortion (beyond statistical fluctuations due to finite sampling) as the theoretical single qubit models exactly replicate the statistics. The quantum distortion is instead due to experimental imperfections in the evolution. We presume that the quantum distortion is non-trivially influenced by systematic error in the evolution due to imperfect optical setup, and that the non-monotonicity is largely a coincidental consequence of this. We have added a sentence noting this distinction between the source of classical and quantum distortion in the figure. We remark that in the original submission, Figure 5(b) erroneously plotted the $N = 4$ data also shown in Figure 5(c); we have now corrected this to show the actual $N = 3$ data.

“In the Discussion, the potential application of the results to quantum clocks is alluded to. Since this topic has seen a lot of traction recently, I think it would be insightful and beneficial to a wide array of readers if the authors could elaborate in a little bit more detail on how they envision their results to potentially impact this field.”

For the avoidance of doubt, we note that by ‘quantum clocks’ we mean the field seeking to understand the fundamental limits of time-keeping from an information theoretic perspective, as opposed to the use of the term as an alternative name for work on atomic clocks. The fields are not entirely unrelated, but we specifically mean the former.

In much of the recent work on quantum clocks, the central object is a ‘clock’ system that within any given time interval Δt produces as output either a ‘tick’, or ‘no tick’ (alternatively, either produces an output, or is silent). In general, the evolution is stochastic, such that one can construct a distribution describing the number of intervals between tick events. The typical goal is to find a means of making the clock as ‘regular’ as possible (e.g., small ratio of standard deviation to mean for the number of intervals between ticks) subject to constraints on the amount of memory (i.e., number of dimensions) available to the clock. It has been shown (e.g., Ref. [26]) that quantum clocks can be more regular than possible with classical clocks of the same dimension. A class of clock systems with particularly favourable properties are ‘reset clocks’, for which the clock system always returns to the same state after a tick, and works on quantum clocks typically restrict their focus to this class. The distribution of ticks for a reset clock constitutes a renewal process. Thus, by being able to implement quantum models of renewal processes, we are in effect able to implement quantum clocks.

We have extended the discussion on this connection to include some such further detail and better clarify what we mean by ‘quantum clock’ in this context.

“It is mentioned in the Discussion that the employed experimental setup can ‘implement single-qubit-memory quantum models of any renewal process’. I might be mistaken, but as far as I can see, there is only a proof of this statement for the renewal process of Eq. (1), but not for all conceivable renewal processes.”

This claim is made with the additional context of the subsequent sentence: “*Whilst not every renewal process can be exactly modelled by a quantum model with a single qubit of memory, recent work has developed techniques for constructing highly-accurate near-exact quantum models of such processes with significantly-reduced memory cost that can be put to use here [5].*”

That is, we do indeed only prove existence of *exact* single-qubit models only for renewal processes of the form Eq. (1) in this manuscript, but we can consider distorted quantum models of arbitrary renewal processes constrained to single-qubit memories, such as those developed in Ref. [5]. These single-qubit distorted models can be implemented in our setup by tweaking only optical equipment that modifies the photon polarisation state, and this is what these sentences are intended to convey. But we can see where the confusion arises, and have modified the sentences to provide better clarity and hopefully avoid giving readers the erroneous impression that any renewal process can be *exactly* simulated with only a single qubit of memory.

“In the Methods, the steady-state distribution of the memory state is used. This distribution should be derived explicitly somewhere for the process at hand.”

It is an oversight that this is not explicitly mentioned. For quantum models of renewal processes the steady state probabilities are proportional to the survival probability $\Phi(n)$. We have now explicitly noted this in the manuscript and provided an appropriate reference for the derivation of this.

“The comparison to classical single-bit-memory models with distortion is very nice and provides a fair comparison between quantum and classical memory requirements. However, I think the discussion of how the bounds for classical models with distortion are obtained (p. 7, last two paragraphs) could benefit from a more detailed explanation. As it stands, there is not enough information provided in the manuscript to allow for an intuitive understanding as to why it is actually possible to bound all distorted classical models in a numerically tractable way. To be clear, I am not asking for a full derivation, just some more high-level

explanation to create a well-rounded picture.”

Indeed, we had been a bit sparing on the details, leaving them to the preprint Ref. [42], as we did not inadvertently wish to usurp priority on the theory behind the approach. We have now added additional details that provide a more fleshed-out intuitive picture of how the approach works. Namely, that: (1) distorted classical models do not benefit from breaking apart the causal states of the ε -machine, and that the optimal approach is to merge the causal states, restricting the mapping from pasts to states of a distorted model to need only consider a mapping from causal states to such states; and (2) models that produce the entire future output one step at a time are a subset of models that produce the entire future at once, which in turn are a subset of models that produce only a finite string of future outputs of length L all at once. By finding a lower bound on the error of the latter, we thus obtain a (not necessarily tight) lower bound on the smallest error achievable in a distorted classical model of the same dimension. While finding the lower bound in this way still bears a complexity that grows exponentially with the number of causal states and L , for the modest $N = [3..8]$ and $L = 2$ considered here, it remains tractable.

“Why can the eigenvalues of ηA_0 be chosen to be 1 and $\exp(i\phi)$ below Eq. (S2) without losing generality? Also, wouldn't this choice then most likely make η complex, while it is assumed to be real throughout the derivation?”

Eq. (S2) can be written as $\eta^N A_0^N = \mathbb{1}$, and thus the eigenvalues λ of ηA_0 must satisfy $\lambda^N = 1$, i.e., $\lambda = \exp(i\phi)$, for $\phi = 2m\pi/N$, where $m \in \mathbb{Z}$. Consider also that the labels of a set of Kraus operators can be considered as the outcomes of measurements on an ancillary system in a particular basis. Provided that the ancilla is (1) not measured in a different basis, and (2) not directly involved in any further evolution, multiplying a Kraus operator by a complex phase factor has no observable physical effect. Thus, we are free to apply a complex phase to A_0 to ensure that η is real and positive, and one of the eigenvalues is 1. We do note though that the second eigenvalue has a bit more freedom, and can be any integer multiple of $2\pi/N$; we have now amended the manuscript to state that we explicitly consider the $m = 1$ case, and included a footnote remarking on the freedom to multiple A_0 by a complex phase.

“While a somewhat trivial step, I think it should be mentioned explicitly at the end of S1 that the two CP maps A_0 and A_1 can indeed always be implemented by means of a unitary that acts on the causal state and a qubit, but do not require a larger environment.”

Done.

“In the second sentence of p. 2, there is a period missing between ‘observations’ and ‘A’.”

Fixed.

We again thank the reviewer for their report. We hope that our above response provides convincing answers to their comments, and thank them for stimulating the associated improvements to our manuscript.

REVIEWERS' COMMENTS

Reviewer #1 (Remarks to the Author):

In the new version of the paper, the authors have revised their manuscript with respect to my comments. The readability of the paper is certainly enhanced and the presentation is fine.

I went through the comments of the other referee too. There were two major issues with the first version of paper: (i) scalability; and (ii) novelty with respect to Ref. [6]. For scalability, I am convinced with the authors that tomography is not needed. However, as also discussed by the authors, there is practical constraint as N increases due to the higher overlap between memory states which may exceed the fidelity of the demonstrated evolution. Considering that we are in NISQ era, I believe that this should not be considered a reason for rejection of the paper.

The novelty issue has also been discussed by the authors and I think there is certainly significant similarities between the idea and the implementation of the experiment in Ref. [6] and the current paper and there are important distinctions, e.g. Markovian vs non-Markovian evolution, too. So, judging the novelty of the paper is indeed a subjective issue. In this issue, I am also inclined towards accepting the paper for Nature Communications.

Based on above, I think the new version of the paper presents important results and, despite overlaps with Ref. [6], contains enough novelty to be published in Nature Communications.

Reviewer #2 (Remarks to the Author):

The authors have addressed my previous comments in their reply and subsequent changes to the manuscript. In particular my concern about the difference between the current work and the results of Ref. [6] has been discussed by the authors in their reply, and I am now convinced that the current manuscript indeed constitutes a significant conceptual and experimental development. Since this had previously been my main concern, I can thus recommend the manuscript for publication.

However, I still think that there are some points that should be addressed. Most prominently, while the authors discussed the difference to Ref. [6] in great detail in their response, the corresponding discussion in the manuscript is in my opinion still lacking. I recognize that the points the authors made in their response are spread throughout the manuscript, but as it stands, this does not create a concrete picture, in what sense the current work is a significant development. Such a discussion that highlights the novelty of the approach with respect to previous ones, would, for example, make sense in the conclusions or when the experimental setup is introduced. Naturally, it does not have to be as detailed as the one provided in the authors' response, but a more concrete comparison with previous results in a single place is required to allow the reader to better embed the provided results in the existing research context.

Likewise, I am not fully convinced by the discussion surrounding the number of time steps ($L=2$), and why this is sufficient to demonstrate that, indeed, non-Markovian statistics can be simulated. One might argue that memory effects only properly come into play when three times are considered, since only then the dependence of the future statistics on the past can actually be observed. I am of course not saying that the authors' experiment is not doing what they claim it does, but the short length of sequences raises the question if the same statistics could not have been achieved in a much simpler way, without storing memory in the environment. In my opinion, the short length of the sequences is not a fundamental problem for the authors' results, but the manuscript would profit from a more detailed discussion of this point.

On a more minor note, I am wondering why it is fundamentally necessary for the number of optical paths to grow exponentially within the employed experimental setup. Naively, it seems easier to

directly perform measurements (as the authors mention at some point), and not to collect the q -sample, thus not requiring an exponential number of optical paths. The same holds true for the question as to why the respective causal state cannot be looped back into the 'beginning' of the experiment. While the authors mention all of these points, in my opinion, it does not become clear what concrete experimental limitations necessitate the way in which the experiment is conducted. I think the paper would gain clarity if these limitations were discussed in some more detail.

On the more pedantic side, I still find the derivation in section S1 highly confusing, since it is barely ever clear what steps follow directly from the previous lines, and what steps are an ansatz. If I am not mistaken, any pair of Kraus operators that create the correct statistics are a proper choice, so any ansatz/guess that will do this is a solution. In this sense, I am fine with the result, but for the individual steps and choices (say, the form of A_1 in Eq. (7)) it is often unclear if they are direct consequences, or simply a choice that turns out to be working nicely in the end. I think a more careful wording of the derivation would make the section much more easily digestible.

Finally, when the scalability of the proposed simulation of classical stochastic processes is mentioned (e.g., line 184) for the first time (I think this happens in a rigorous sense for the first time in line 184), I think it would be fair to be explicit about the fact that this scalability comes with the caveat that at some point the overlap of the respective causal states becomes 'too large'. This, then leads to large distortions that counter the reduction in memory dimension. While the authors claimed that this is a practical rather than a fundamental limit, I personally fail to see the difference between the two, since a lack of perfect implementation makes the desired scaling fundamentally unattainable for high N . Independent of this question of practical vs. fundamental, the caveat this advantage comes with should at least be alluded to when introduced. I do recognize that it is mentioned later on, but I think an immediate mentioning of this limitation would help the reader grasp directly that there are limits to this nice scalability.

As mentioned above, I am now convinced that this manuscript constitutes a significant step in the development of quantum technologies for the simulation of complex processes. The points I raised above are rather minor, and I thus recommend this work for publication.

Response to Reviewers

We thank the reviewers for their further feedback on our manuscript, and are happy to see that they both now support publication of the manuscript. We thank them for their role in catalysing improvements to the presentation of our results in the manuscript.

We here respond to the further minor comments provided by Reviewer 2 in their second review.

“Most prominently, while the authors discussed the difference to Ref. [6] in great detail in their response, the corresponding discussion in the manuscript is in my opinion still lacking. I recognize that the points the authors made in their response are spread throughout the manuscript, but as it stands, this does not create a concrete picture, in what sense the current work is a significant development. Such a discussion that highlights the novelty of the approach with respect to previous ones, would, for example, make sense in the conclusions or when the experimental setup is introduced. Naturally, it does not have to be as detailed as the one provided in the authors’ response, but a more concrete comparison with previous results in a single place is required to allow the reader to better embed the provided results in the existing research context.”

We have now incorporated further details from the discussion in our previous response into the manuscript. Specifically, we have now included a condensed discussion of the key advancements our manuscript makes over Ref. [6] at the end of our experimental details subsection.

“Likewise, I am not fully convinced by the discussion surrounding the number of time steps ($L = 2$), and why this is sufficient to demonstrate that, indeed, non-Markovian statistics can be simulated. One might argue that memory effects only properly come into play when three times are considered, since only then the dependence of the future statistics on the past can actually be observed. I am of course not saying that the authors’ experiment is not doing what they claim it does, but the short length of sequences raises the question if the same statistics could not have been achieved in a much simpler way, without storing memory in the environment. In my opinion, the short length of the sequences is not a fundamental problem for the authors’ results, but the manuscript would profit from a more detailed discussion of this point.”

We agree that if one is only presented with measurement statistics, and no other contextual details, one would need three timesteps to verify that the observed statistics are non-Markovian. However, in this case we do have additional context to the statistics observed – namely, in the preparation of the initial memory state. In this case, we see that the two-time statistics observed do change based on the initial memory state – and crucially, the conditional statistics of the second measurement given the first varies with the initial memory state. Since the setup performing the evolution is not changed based on the choice of initial memory state, it must be the state of the memory propagated between the two timesteps that is responsible for the changing conditional statistics, and conversely, such changing conditional statistics herald the presence of memory. We have expanded our discussion in the manuscript on this point.

“On a more minor note, I am wondering why it is fundamentally necessary for the number of optical paths to grow exponentially within the employed experimental setup. Naively, it seems easier to directly perform measurements (as the authors mention at some point), and not to collect the q-sample, thus not requiring an exponential number of optical paths. The same holds true for the question as to why the respective causal state cannot be looped back into the ‘beginning’ of the experiment. While the authors mention all of these points, in my opinion, it does not become clear what concrete experimental limitations necessitate the way in which the experiment is conducted. I think the paper would gain clarity if these limitations were discussed in some more detail.”

For the purpose of simulating the statistics, it is not fundamentally necessary for the number of optical paths to grow exponentially, and we hope that we have not given this impression. It is only fundamentally necessary to do this if one is also intending to generate the full q-sample. However, it is a

choice of our particular setup to use additional paths rather than looping back at each timestep; this latter alternative approach would carry other practical drawbacks that we sought to avoid, in terms of requiring nondeterministic optical operations. We have refined our discussion of the experimental details to provide further clarity in this regard.

“On the more pedantic side, I still find the derivation in section S1 highly confusing, since it is barely ever clear what steps follow directly from the previous lines, and what steps are an ansatz. If I am not mistaken, any pair of Kraus operators that create the correct statistics are a proper choice, so any ansatz/guess that will do this is a solution. In this sense, I am fine with the result, but for the individual steps and choices (say, the form of A_1 in Eq. (7)) it is often unclear if they are direct consequences, or simply a choice that turns out to be working nicely in the end. I think a more careful wording of the derivation would make the section much more easily digestible.”

The reviewer is correct that any pair of Kraus operators that lead to the correct statistics are a valid choice, and that this is not a unique choice. We have now noted this in S1, and augmented our derivation of our particular choice to better emphasise which steps would have allowed for alternative options in their progression.

“Finally, when the scalability of the proposed simulation of classical stochastic processes is mentioned (e.g., line 184) for the first time (I think this happens in a rigorous sense for the first time in line 184), I think it would be fair to be explicit about the fact that this scalability comes with the caveat that at some point the overlap of the respective causal states becomes ‘too large’. This, then leads to large distortions that counter the reduction in memory dimension. While the authors claimed that this is a practical rather than a fundamental limit, I personally fail to see the difference between the two, since a lack of perfect implementation makes the desired scaling fundamentally unattainable for high N . Independent of this question of practical vs. fundamental, the caveat this advantage comes with should at least be alluded to when introduced. I do recognize that it is mentioned later on, but I think an immediate mentioning of this limitation would help the reader grasp directly that there are limits to this nice scalability”

We agree with the reviewer that – independent of the question of practical versus fundamental limits – practical limitations are an important consideration. We have now made mention of this caveat in line 184, i.e., that the theoretical scaling advantage is in practice tempered by a trade-off with the precision of the implementation in resolving and manipulating states with high overlap.